# Wafer Defect Image Generation Method Based on Improved Styleganv3 Network

**DOI:** 10.3390/mi16080844

**Published:** 2025-07-23

**Authors:** Jialin Zou, Hongcheng Wang, Jiajin Zhong

**Affiliations:** 1School of Electrical Engineering and Intelligentization, Dongguan University of Technology, Dongguan 523808, China; 15843173336@163.com; 2School of Computer Science and Technology, Dongguan University of Technology, Dongguan 523808, China; jiajinzhong1128@163.com

**Keywords:** Generative Adversarial Network, wafer defect generation, deep learning

## Abstract

This paper takes a look at training a generator model based on a limited dataset that can fit the distribution of the original dataset, improving the reconstruction ability of wafer datasets. High-fidelity wafer defect image generation remains challenging due to limited real data and poor physical authenticity of existing methods. We propose an enhanced StyleGANv3 framework with two key innovations: (1) a Heterogeneous Kernel Fusion Unit (HKFU) enabling multi-scale defect feature refinement via spatiotemporal attention and dynamic gating; (2) a Dynamic Adaptive Attention Module (DAAM) adaptively boosting discriminator sensitivity. Experiments on Mixtype-WM38 and MIR-WM811K datasets demonstrate state-of-the-art performance, achieving FID scores of 25.20 and 28.70 alongside SDS values of 36.00 and 35.45. The proposed method in this article helps alleviate the problem of limited datasets and makes an important contribution to data preparation for downstream classification and detection tasks.

## 1. Introduction

The quantity and quality of wafer defect maps are crucial for effective wafer defect detection. In semiconductor wafer manufacturing, the value of each wafer can reach thousands of dollars, and even minor defects can result in the entire wafer being scrapped. This underscores the significance of wafer defect maps, as the accurate detection of wafer defects is paramount.

Wafer defect maps record the location, type, and distribution of defects on the wafer surface, serving as the core basis for defect detection by enabling a systematic analysis of defect patterns. Complex defects, such as mixed-type defects or those with submicron sizes, involve intricate morphological features that traditional optical tools fail to classify—these defects often indicate specific failures in manufacturing stages like photolithography or etching. Early manual methods relied on visual inspection under microscopes, which was error-prone and inefficient.

Traditional optical inspection equipment can identify wafer defects; however, it struggles with complex defect classification and root cause analysis. In contrast, early manual detection methods were characterized by low efficiency and accuracy. With advancements in artificial intelligence, machine learning-based approaches have significantly improved defect detection accuracy, although they still encounter certain limitations. For instance, Tao et al. [1] proposed a Bayesian inference-based clustering algorithm to identify spatial defect patterns. Additionally, Chao et al. [2] introduced a novel multi-class Support Vector Machine (SVM) identification system that incorporates a defect clustering index. Furthermore, Jessnor et al. [3] evaluated four machine learning classifiers—k-Nearest Neighbors (k-NN), logistic regression, Stochastic Gradient Descent (SGD), and SVM—for wafer defect detection, all of which were validated through experimental results.

Subsequently, deep learning-based methods have supplanted traditional machine learning techniques for wafer defect recognition. However, all methodologies necessitate high-quality, large-scale datasets to ensure optimal model performance. Wafer defect map generation technology simulates defect samples from limited real data to facilitate algorithm training. To tackle the authenticity and adaptability challenges present in current methods, this paper enhances StyleGANv3 [4] and introduces a generation algorithm that incorporates process parameters to improve semiconductor quality control and optimize costs.

The main contributions of this article are as follows:

(1) The generator incorporates a Heterogeneous Kernel Fusion Unit (HKFU), which employs an adaptive weight scaling design that does not utilize batch normalization. By integrating multi-scale channel–spatial attention with temporal residual modules, and leveraging a dynamic gating mechanism for adaptive multi-scale feature selection and decoupling enhancement, the HKFU significantly enhances the physical authenticity of the generated defect maps, improves multi-scale feature representation, and stabilizes the training process.

(2) The Dynamic Adaptive Attention Module (DAAM) is designed for the discriminator. By leveraging learnable basis weights and linear interpolation, DAAM integrates channel–space dual-path attention to dynamically optimize feature extraction with minimal computational cost. This approach enhances sensitivity to fine-grained defect features and improves the quality of generated images.

(3) Experiments were conducted using the MIR-WM811K dataset [5] and the Mixtype-WM38 wafer defect map dataset [6]. The effectiveness of the proposed model was evaluated through FID [7] and SDS [8] metrics across different datasets.

## 2. Related Works

Existing approaches either focus on improving detection accuracy through network optimization or utilize generative models for data augmentation, but they lack the integration of defect map generation with deep learning-based detection to address complex defect classification and root cause analysis challenges—these limitations underscore the necessity of the present study.

### 2.1. Wafer Defect Detection Using Deep Learning Methods

Deep learning-based wafer defect detection has gained significant traction in recent years. Wang et al. [9] proposed a novel method called CR-NMS, which utilizes Coverage Percentage (CoP) instead of Intersection over Union (IoU) for bounding box regression within a two-stage algorithm. This approach enhances detection by considering the correlations among defects. Jariya et al. [10] combined convolutional neural networks (CNNs) and deep neural networks (DNNs) for defect classification, achieving improved accuracy through multi-layer convolution and the perceptual capabilities of DNNs. Francisco et al. [11] introduced a two-stage method that integrates traditional computer vision techniques with the lightweight SqueezeNet architecture. This method employs geometric data augmentation and grid search for hyperparameter optimization in defect detection and classification.

Chen et al. [12] proposed a deep convolutional neural network (CNN) that integrates an improved Convolutional Block Attention Module (I-CBAM), multi-branch attention mechanisms, a ResNeXt50 backbone, and an Error-Correcting Output Code Support Vector Machine (ECOC-SVM) classifier to enhance the extraction and classification of defect features. Zheng et al. [13] developed a background subtraction-based method that incorporates an improved Faster R-CNN, utilizing spectral analysis for background reconstruction, spatial attention, and K-means clustering for anchor initialization to mitigate interference. Zhang et al. [14] introduced the Multi-Level Relay Vision Transformer (MLR-WM ViT), which combines convolutional additive self-attention with a relay classifier to investigate defect relationships, thereby improving detection accuracy.

### 2.2. Generation Methods in Industrial Defect Detection

Davide et al. [15] proposed a deep learning (DL)-based data augmentation method that utilizes GAN-generated semantic layouts in conjunction with texture synthesis to expand datasets and enhance the recognition of low-cardinality categories. Yun et al. [16] developed a metal defect classification algorithm using a Conditional Continuous Variational Autoencoder–deep convolutional neural network (CCVAE-DCNN), addressing data imbalance by generating additional samples through CCVAE. Niu et al. [17] introduced SDGAN, which employs two diversity discriminators and cyclic consistency loss to generate high-quality defect images from defect-free data, thereby improving deep learning recognition. Zhao et al. [18] proposed Diff-Augment, which applies differentiable augmentations to GAN samples to stabilize training and mitigate discriminator overfitting. Tseng et al. [19] presented a GAN regularization method tailored for limited data, linking the objective to LeCam divergence through a novel term to enhance generalization and training stability.

Zhang et al. [20] proposed the Multi-Scale Generative Adversarial Network (MAS-GAN), which incorporates self-attention mechanisms and non-leakage data augmentation techniques to address the challenge of insufficient industrial surface defect samples and to enhance the quality of generated images. Xu et al. [21] developed a small-sample road crack detection method that utilizes Generative Adversarial Networks (GANs) for data expansion and convolutional neural networks (CNNs) with transfer learning, integrating Class Activation Mapping (CAM) for visual evaluation. Deng et al. [22] employed DG2GAN, which features cyclic consistency loss, a Jensen–Shannon divergence-optimized discriminator loss, and DG2 adversarial loss to generate high-quality, diverse defect images from defect-free data. Wang et al. [23] introduced a multi-scale Inpainting GAN framework that utilizes multi-scale masks and gradient amplitude similarity loss to facilitate surface defect detection through image restoration. Zhou et al. [24] utilized deep convolutional GANs (DCGANs) to generate images of steel plate defects, optimizing the model with batch normalization and various activation functions to improve recognition accuracy.

## 3. Proposed Methods

### 3.1. Heterogeneous Kernel Fusion Unit

The Heterogeneous Kernel Fusion Unit (HKFU) is characterized by a symmetric encoder–decoder architecture that incorporates three levels of downsampling and upsampling paths. This design facilitates cross-level feature fusion through a dynamic gating mechanism. Furthermore, it innovatively integrates multi-scale convolutional kernels, spatiotemporal attention, and an adaptive gating system to achieve end-to-end feature refinement. Adhering to a ‘shrinking–refining–expanding’ design principle, the HKFU ensures the fusion of multi-scale information while maintaining the integrity of the feature space, as illustrated in Figure 1. This symmetric encoder–decoder design is distinct from traditional single-path fusion architectures, as it explicitly preserves feature resolution consistency across scales. The dynamic gating mechanism, unlike fixed-weight fusion methods, adaptively prioritizes informative features while suppressing noise, a key innovation for handling the high variability of wafer defect patterns. Such a design can be readily extended to other high-precision manufacturing inspection tasks where multi-scale feature alignment is essential.

The multi-scale attention (MSA) framework was developed based on the principles of multi-scale visual perception [25]. It employs three parallel convolution branches with kernel sizes of 3×3, 5×5, and 7×7 to effectively extract features corresponding to local textures, medium structures, and global semantics. The framework incorporates dual-channel attention mechanisms: channel attention mitigates irrelevant features through global pooling, while spatial attention accentuates defect edges using max–mean pooling. Additionally, temporal residual connections are integrated to alleviate gradient decay, thereby establishing a ‘memory-update’ pathway for cross-layer feature fusion. This architecture significantly improves the representation and generation accuracy of multi-scale defects in wafer maps, as illustrated in Figure 2. The choice of 3×3, 5×5, and 7×7 kernels is tailored to wafer defect characteristics: 3×3 captures fine textures, 5×5 targets medium defects, and 7×7 covers large-area anomalies. Unlike single-attention mechanisms, the dual-channel design simultaneously filters redundant channels and enhances spatial saliency, a novelty that addresses the “signal-to-noise imbalance” in wafer maps. Here, the temporal residual connections, originally developed for sequential data, enable the model to “remember” prior defect patterns, making it adaptable to time-series wafer inspection.

Dynamic feature gating is applied to the cross-layer connections of the HKFU architecture, adaptively weighting the encoder and decoder features through a learnable gating mechanism to address information imbalance. The HKFU architecture preserves the correspondence between encoder and decoder layers, integrating multi-scale attention modules after each downsampling step. The final attention layer employs Sigmoid activation for spatial weighting, thereby forming residual connections. This module replaces batch normalization with adaptive weight scaling to eliminate dependence on batch statistics, effectively covering most image receptive fields. The learnable gating mechanism dynamically adjusts weights based on feature importance—for instance, assigning higher weights to encoder features containing defect location cues and decoder features with texture details, thus resolving the information dominance issue common in symmetric architectures. Replacing batch normalization with adaptive scaling eliminates batch-dependent statistical bias, a critical improvement for small-batch wafer data. This design not only boosts generator accuracy for wafer maps but also offers a general solution for generative models in data-scarce industrial scenarios. Incorporating HKFU into the generator significantly enhances model accuracy, as illustrated in Figure 3.

### 3.2. Dynamic Adaptive Attention Module

The Dynamic Adaptive Attention Module (DAAM) incorporates an innovative design characterized by dual-path attention fusion and dynamic training perception [26]. This approach addresses the critical issue of discriminator capability imbalance commonly encountered in traditional GAN training. Figure 4 illustrates the structural diagram of the DAAM model.

DAAM primarily consists of spatial and channel attention branches. The spatial branch extracts essential spatial information through channel-wise mean and maximum value maps. The mean map computes the arithmetic mean of all channels at each spatial position using global average pooling, resulting in a smooth regional energy representation. In contrast, the max map captures local salient feature points by identifying extreme values along the channel dimension. These two maps are concatenated and fed into a 7 × 7 large kernel convolutional layer for spatial context modeling, ultimately generating a spatial weight map through Sigmoid activation. The channel branch compresses the two-dimensional features of each channel into scalar descriptors via adaptive global average pooling, creating a channel-level global context representation. This representation then undergoes a non-linear transformation through a bottleneck structure consisting of two 1 × 1 convolutions, which first reduces the dimensionality with an 8:1 compression ratio and subsequently restores the original dimension. Finally, channel weight vectors are generated via Sigmoid activation.

During training, the gamma parameter adaptively adjusts the discriminator’s attention strength. Initially, a smaller gamma reduces the attention to prevent excessively strong discrimination. As training progresses, gamma linearly interpolates to 1.0, thereby enhancing attention for improved detail discrimination.(1)$$\gamma \left(t\right)=\gamma_{0}\left(1-\frac{t}{T}\right)+\frac{t}{T}$$

Formula (Equation 1) is a linear interpolation formula. Among them, $\gamma_{0}$ represents the initial value, which is preset to 0.1 here. *T* is the preset total training round, and *t* is the current training round. This design achieves smooth transitions through linear interpolation to ensure training stability. DAAM is mainly used for the discriminator, and the structure is shown in Figure 5.

## 4. Experiments

In this section, we will compare the performance of our model with other industrial defect image generation models using our wafer defect dataset. We will evaluate the metrics of these models across various datasets.

### 4.1. Datasets and Experimental Settings

This study utilizes the MIR-WM811K (comprising 2249 images) [5] and Mixtype-WM38 (comprising 1760 images) [6] datasets for few-sample learning experiments, which are motivated by the constraints posed by limited defect samples and equipment availability. All images are adjusted to 128 × 128 pixels to meet the resolution requirements of the model (the original resolution was 256). Both datasets encompass eight common categories, including center, donut, edge-loc, edge-ring, loc, near-full, random, and scratch. To optimize model performance while managing training costs, data from all categories are uniformly mixed into a single training set, thereby evaluating the model’s capability to adapt to the dataset’s distribution.

As shown in Figure 6, the partial display of the wafer dataset in this article consists of eight categories, with one image selected for each category. The category to which each image belongs is located below each image.

In this section, Pytorch is used as the basic framework for deep learning, with a CPU of 12 vCPU Intel (R) Xeon (R) Platinum 8352V CPU @ 2.10 GHz and a GPU of NVIDIA GeForce RTX 4080 Super 32 GB. For fair comparison, all models in this article were tested based on this environment, and each model was trained using the best parameters according to its own paper. The output image size of the model is 128 × 128, the training iteration is 25,000 times (to fully train the model), the batch is set to 48, and the gamma parameter in this article is set to 8.

### 4.2. Evaluation Metrics

FID [7] employs pre-trained Inception networks to extract high-level image features, calculates the mean and covariance differences between the feature distributions of real and generated images, and assesses their distribution matching. A smaller FID value signifies that the authenticity and diversity of the generated images are more closely aligned with the real data distribution, establishing it as a widely accepted objective metric for evaluating the performance of generative models. The calculation of FID is presented in Formula (Equation 2).(2)$$\mathrm{FID}={∥\mu_{\mathrm{real}}-\mu_{\mathrm{fake}}∥}^{2}+Tr\left(\Sigma_{\mathrm{real}}+\Sigma_{\mathrm{fake}}-2{\left(\Sigma_{\mathrm{real}}\cdot \Sigma_{\mathrm{fake}}\right)}^{1/2}\right)$$

In the formula, μ and Σ represent the mean and covariance matrices of the real image and the generated image, respectively, which are statistical measures for the two distributions. *Tr* represents the trace of the matrix, which comprehensively measures the differences in mean and covariance structure.

This article also employs the Surface Defect Score (SDS) [8] as an evaluation metric to provide a more authoritative assessment of the model’s effectiveness. The SDS is calculated by mixing real images with fake images (labeled as 1 and 0, respectively) generated by the model. The ResNet18 [27] binary classification model is then trained using both fake and real images. Subsequently, this model is utilized to classify the remaining fake images, allowing for the evaluation of the generated images’ quality based on classification accuracy. The calculation formula for SDS is presented in Formula (Equation 3). A higher SDS indicates that the generated image is closer to the distribution of real images.(3)$$\mathrm{SDS}=100\%-\frac{1}{n}\sum_{x=1}^{n}\mathrm{Resnet}18\left(G\left(x\right)\right)$$

### 4.3. Comparative Experiment

In this section, we mainly describe the performance comparison of our model with other classic and excellent defect generation models on wafer datasets such as Mixtype-WM38 and MIR-WM811K. As mentioned earlier, the evaluation indicators used include FID and SDS. The comparative models discussed in this article include GAN [28], LSGAN [29], WGAN [30], RSGAN [31], and StyleGANv3.

Table 1 compares the FID performance of our model with that of other baselines across different datasets. On Mixtype-WM38, all models exhibit FID scores below 40, indicating a good fit. Notably, our model achieves the lowest FID score on this dataset, demonstrating superior generalization ability. Since FID quantifies the distance between the feature distributions of generated and real data using Inception-v3 embeddings, a lower score indicates a closer match in both global structure and fine-grained details between synthetic and authentic samples. This tight alignment means that the model has learned not just superficial patterns but the underlying statistical properties of the dataset, enabling it to generate samples that generalize beyond visible examples and remain consistent with real-world defect characteristics. The generated data at a resolution of 128 × 128 best aligns with the original data distribution.

In the MIR-WM811K dataset, only three models achieved a FID below 40, while the others exceeded this threshold. This discrepancy may be attributed to the inferior original data quality of MIR-WM811K compared to Mixtype-WM38, which adversely affects feature fitting. Notably, our model once again attained the lowest FID, demonstrating substantial original data fitting despite only marginal improvements in distribution effects. Figure 7 and Figure 8 illustrate the FID variation curves of the different models during training on the two datasets, respectively.

Table 2 presents the SDS metrics of various models in Mixtype-WM38 and MIR-WM811K. The model proposed in this paper achieved the highest SDS scores across both datasets, demonstrating superior fit compared to other comparative models in terms of generalization and robustness. This result strongly validates the effectiveness of the proposed method.

### 4.4. Comparison of Images Generated by Different Models

In this section, we primarily present comparative graphs of wafer generation data to illustrate the effectiveness of our model in actual image generation. Figure 9 and Figure 10 display the generated graphs of different models on the Mixtype-WM38 dataset and the MIR-WM811K dataset, respectively.

In the generated graphs of the two datasets, the first row displays the data produced by the GAN, which presents an overall outline but appears slightly blurred. The second row illustrates the graph generated by the LSGAN, demonstrating noticeable improvements compared to the GAN output. The third row features the generated image from the WGAN, which exhibits a similar quality to that of LSGAN. The fourth row presents the generated image from the RSGAN, showing a marginal enhancement in the results. The fifth row showcases the generated image from StyleGANv3, which further improves the output quality. The sixth row depicts the generated graph from the model discussed in this article, exhibiting a data distribution that is closer to that of the original dataset. Finally, the last row contains the original image data, serving as a benchmark for comparing the quality of the generated images.

Specifically, our model’s outputs (sixth row) exhibit more consistent defect edges, finer texture details, and closer alignment with the real defect distribution (last row) compared to other models—for instance, clearer boundary definition than GAN/LSGAN, more stable defect morphology than WGAN/RSGAN, and better preservation of subtle defect features than StyleGANv3. This category-specific, feature-matched comparison highlights that our model not only achieves quantitative superiority in metrics but also generates qualitatively more realistic images with faithful defect characteristics.

## 5. Conclusions

This work presents a novel wafer defect generation method by enhancing StyleGANv3 with HKFU and DAAM modules. HKFU’s multi-scale attention and gating mechanism improve physical fidelity and training stability, while DAAM dynamically optimizes feature discrimination. Our model achieves FID values of 25.20 (Mixtype-WM38) and 28.70 (MIR-WM811K), with SDS values of 36.00 and 35.45, confirming superior generation quality. The framework effectively addresses data scarcity in industrial defect detection, offering practical value for semiconductor manufacturing.

These high-fidelity synthetic defect maps directly address the critical challenge of limited real-world defect samples in semiconductor manufacturing. By augmenting training datasets with diverse synthetic samples, this enhances the robustness of downstream defect detection models—particularly for rare defect types—thereby improving detection accuracy and reducing false negatives. This translates to more reliable quality control in wafer production, enabling earlier identification of manufacturing anomalies, minimizing scrap rates, and ultimately driving cost savings in high-value semiconductor manufacturing processes.

## Figures and Tables

**Figure 1 micromachines-16-00844-f001:**
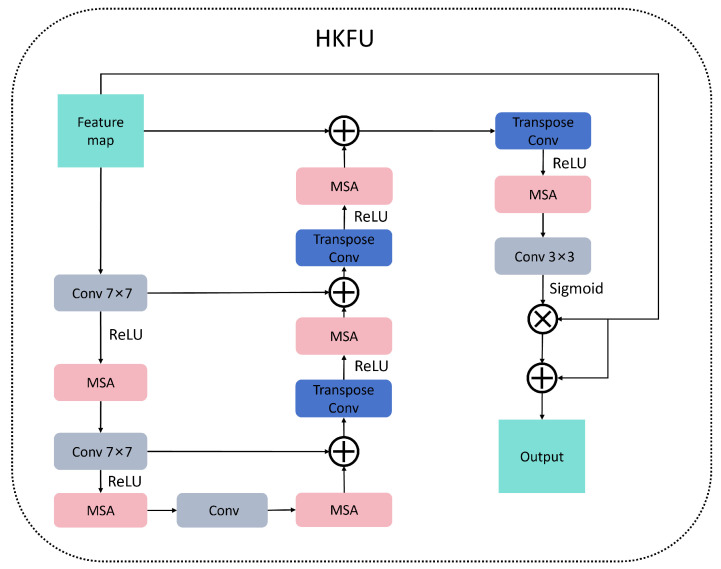
HKFU model architecture.

**Figure 2 micromachines-16-00844-f002:**
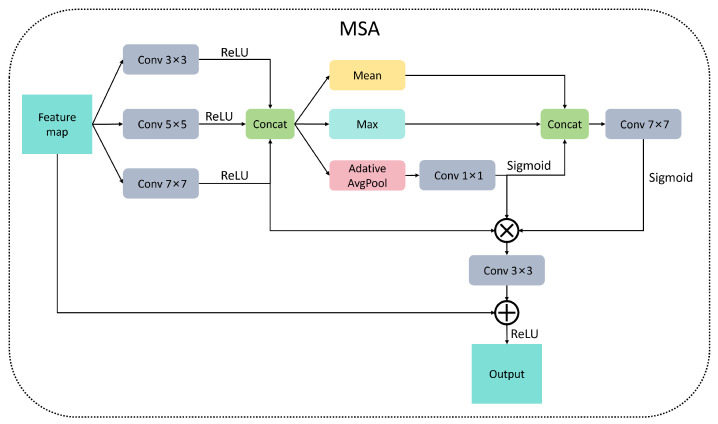
MSA model architecture.

**Figure 3 micromachines-16-00844-f003:**
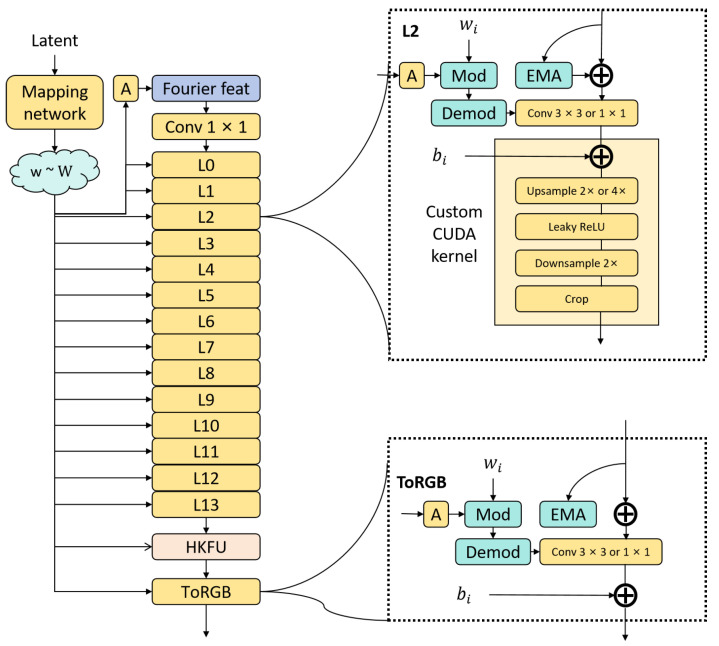
Generator architecture.

**Figure 4 micromachines-16-00844-f004:**
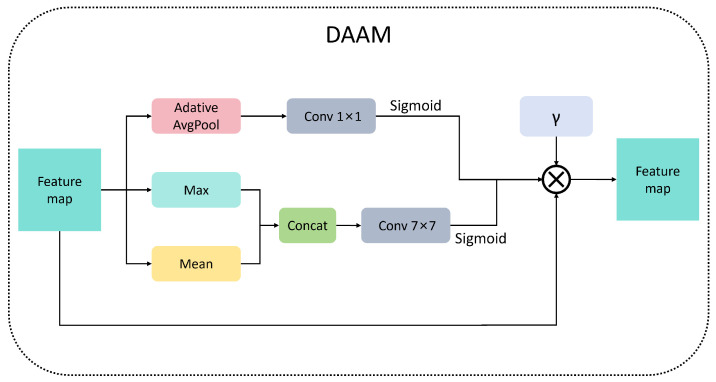
DAAM model architecture.

**Figure 5 micromachines-16-00844-f005:**
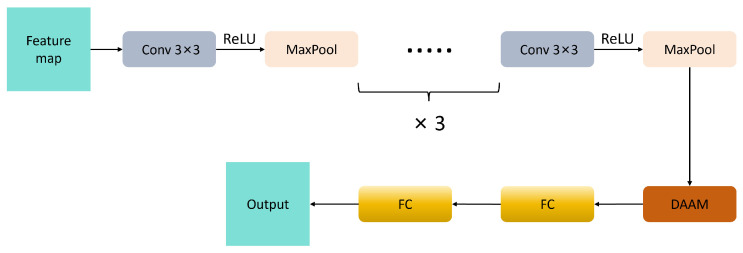
Discriminator architecture.

**Figure 6 micromachines-16-00844-f006:**
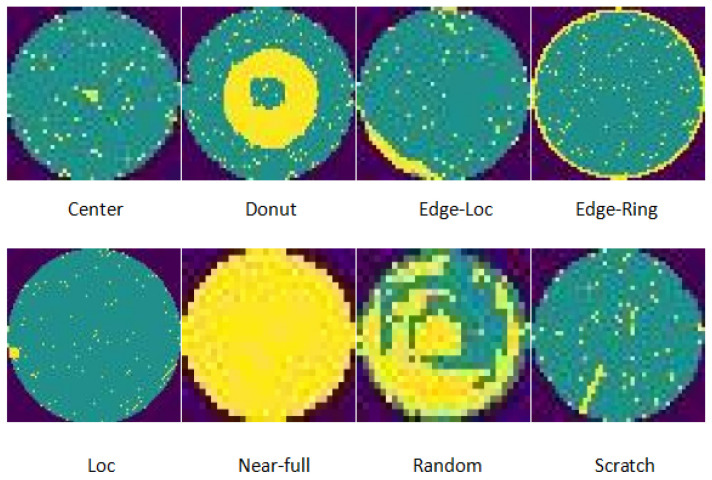
Original data.

**Figure 7 micromachines-16-00844-f007:**
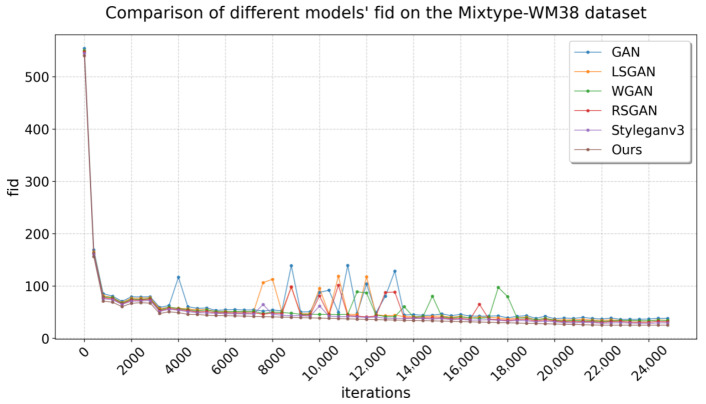
FID on the MixType-WM38.

**Figure 8 micromachines-16-00844-f008:**
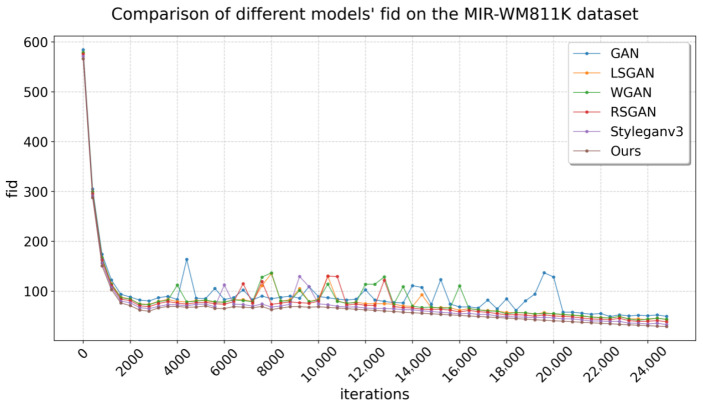
FID on the MIR-WM811K.

**Figure 9 micromachines-16-00844-f009:**
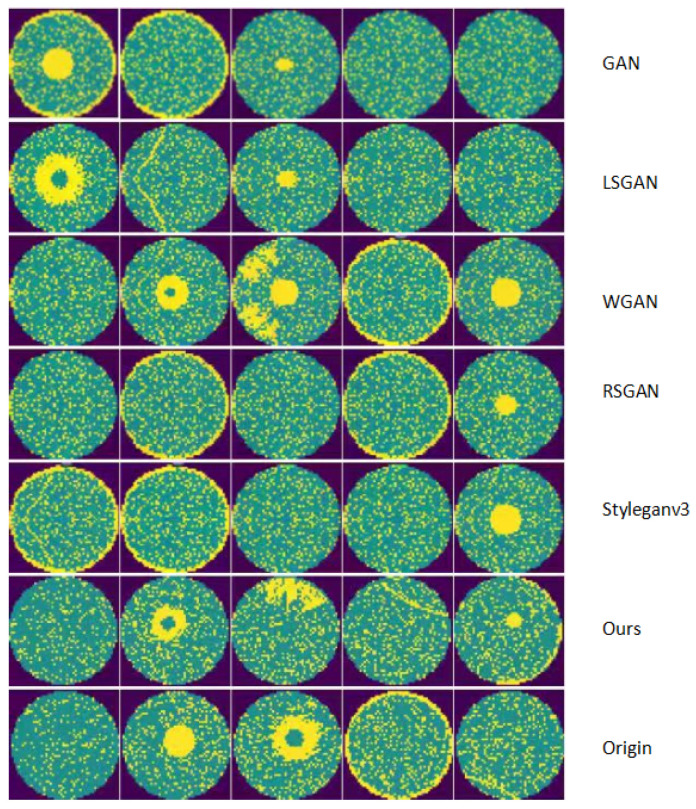
Mixtype-WM38-generated image.

**Figure 10 micromachines-16-00844-f010:**
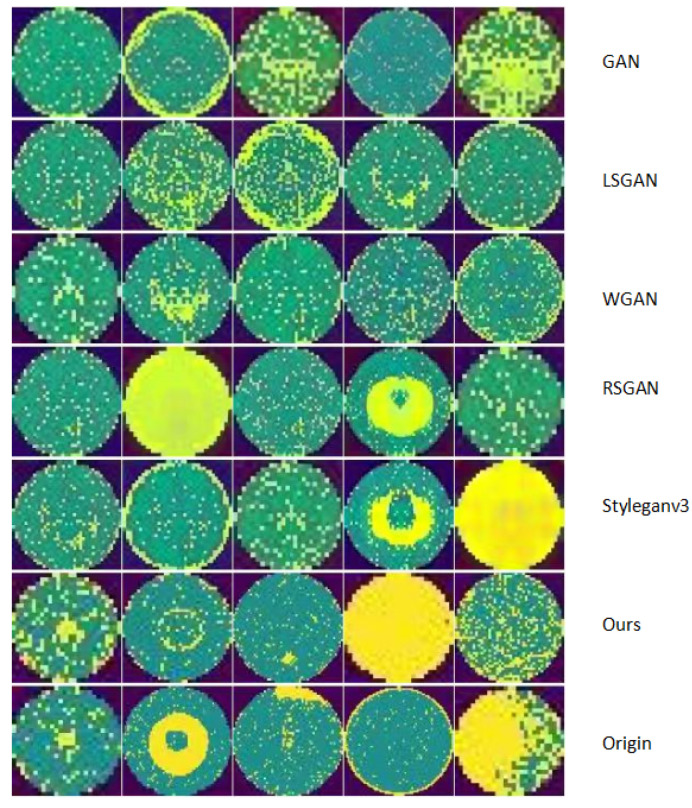
MIR-WM811K-generated image.

**Table 1 micromachines-16-00844-t001:** Comparison of FID indicators.

Models	FID in Mixtype-WM38	FID in MIR-WM811K
GAN	36.71	50.17
LSGAN	33.62	42.80
WGAN	34.17	42.31
RSGAN	31.22	38.01
Styleganv3	28.86	33.91
Ours	25.20	28.70

**Table 2 micromachines-16-00844-t002:** Comparison of SDS indicators.

Models	SDS in Mixtype-WM38	SDS in MIR-WM811K
GAN	8.00	6.12
LSGAN	16.32	14.37
WGAN	16.88	15.68
RSGAN	18.16	18.00
Styleganv3	22.36	21.33
Ours	36.00	35.45

## Data Availability

The raw data supporting the conclusions of this article will be made available by the authors upon request.

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
