# Peer review of "Wafer Defect Image Generation Method Based on Improved Styleganv3 Network"

_micromachines, 2025, doi:10.3390/mi16080844_

Round 1

Reviewer 1 Report

Comments and Suggestions for Authors

abstract:

high fidelity wafer defect image generation. limited real data, poor physical authenticity of existing models. how is your model more physically authentic? you rely on the same datasets, and your generative model is not physically based.

new model architecture with targeted components to improve feature quality (HKFU, DAAM). improved performance in reconstruction of wafer defect datasets.

significantly advances semiconductor defect inspection. it is not clear how you support linking image generation to improved defect inspection? how does an improved model for generating realistic defect images help with real-world wafer inspection?

intro:

[p1 l13] you just straight into wafer defect maps without providing much background or context. can you provide a bit more relevant context for your domain and industry. what are these wafers used for? also the link between defect maps and defect detection is not clearly explained. how does having a defect map enable detection?

[p1 l19] can you define what complex defects are, and why they cannot be classified with traditional optical inspection. and how does this enable root cause analysis? are specific defects indicative of manufacturing stage faults? and you contrast to early manual methods, being poor performance, but what are these? are they not optical methods? it is not clear the distinction between characterisation or image acquisition and analysis method. please clarify these statements

[p2 l34] you outline deep learning models for generation somehow enhance defect detection models but is not clear or explicit. you propose to generate synthetic datasets at sufficient scale to enable improved training of deep models for detection? and you link this directly to quality control and cost optimistation, but don't motivate this connection. can you provide more detail on how these developments will actually practically improve quality control?

overall the introduction covers most points, but is lacking in context and clarity, and additional detail for the benefit of the reader. can you improve based on the above comments, as this will add clear value to the manuscript.

related work:

overall a respectable coverage of related literature work. however there is no real synthesis of this information. can you summarise the outstanding gaps from this survey that motivate the need for your proposed contributions? what specific limitations exist within these methods for your application? in what way do they limit the application of deep ml models for defect map generation and detection?

methods:

[p4] overall the detail of developed architectural enhancements are adequate, with good illustrative diagrams of each. however for the benefit of the reader, more detail could be included in the figure captions of each F1/2/3. this is an ideal space to provide detail and context for the figure contents within the wider model architecture

[p3-5] it is important to highlight the specific novel aspects of your contributions, as it is not immediately clear. you present model component architectures specific to your problem domain, but how are these fundamentally novel ideas.  it is hinted at throughout your description and discussion, but more emphasis on specific novel aspects or ideas can help guide the reader. specifically, how do these insights in design for enhanced performance here enable utilisation for other tasks or domains? perhaps more clear explanation of why these design choices were made in contrast to alternative options?

experiments:

[p6 l170] you state images are downscaled to 128^2, but not what the original resolution was? and does such a severe scaling impact the application of your model in a practical sense? can you provide context to the typical image acquisistion details, and requirements for adequate defect detection. ie what resolution is required for defect maps to train models and ensure robust defect detection? 128x128 sounds very small. is it possible to increase the increase the input resolution of your model? would this be computationally costly?

[p6 l171] you list common defect labels, but completely without any context, details, or examples. can you provide some examples of specific defect types and describe their unique characteristics. to those unfamiliar with these specific datasets or domain, it is not obvious what these class labels indicate

[p6 l176] while not so important, but given you provide CPU architecture but only GPU vram, can you also include GPU model. the vram is indicative of the GPU performance tier, but it is useful to provide computational power capacity also

[p6 l179] you state a fixed number of iterations for all model training. why was this number chosen? could you not implement some form of automated assessment of performance for early-stopping or equivalence?

[p7 l204] you list a series of comparison models, but provide no motivation for the choices in this list.  can you expand with a little more detail on the reference model selection?

[p7 l207] please clarify why FID below 40 indicates a good fit. can you provide a reference to support this statement?

[p7 l208] you state best-in-class performance of your model demonstrates generalisation ability. how so? generalisation in what context. you are simply testing performance against the datasets used for training.  suggest be more clear on your statement, as these results may not generalise to significantly different datasets

[p8] both figures F6/7 could be scaled to increase axes and legend font size to enhance readability

[p9] both figures F8/9 are impossible to interpret without labels or axes. this needs to be fixed. while you explain them in the body text, there isn't even any detail in the figure captions. please update both figures with row/columnn labels to clearly indicate model and sample

[p10 l237] given this is an unconditioned generative model, how are these images produced for comparison to ground truth? it is not clear if you have simple chosen random images that look comparable. are they grouped by defect class? I agree there is reasonable qualitative alignment between real dataset images and those generated. however surely the columns should be chosen as to compare images featuring comparable defect features.  can you update these figures to enable such a like-for-like comparison? as it stands, it is not clear from these figures that your model produces a better image than any of the others.  as such these figures provide no value to the manuscript and need to be improved. it is also worth highlighting clear specific examples of comparison between models for discussion.  this should highlight the strength and limitations of each model, and how they align to the ground truth images

conclusion:

[p10 l244] you can improve this final statement by providing a more clear link between your proposed model and the outcomes.  you have developed a model that enables higher quality defect map image generation.  how does this lead to practical value for the semiconductor manufacturing industry? support this statement with the steps in between to make clear the the reader the link to application outcomes

summary:

overall the manuscript is reasonably well structures and written, with clear methods and results showing good outcomes.  however a number of areas need improvement, including clarification of context and links to outcomes.  and figures need to be fixed based on comments outlined above.

Author Response

Please see in attachment

Reviewer 2 Report

Comments and Suggestions for Authors

This paper presents an Image Generation Method for wafer defect which can achieve good performance.

  1. For the generated images, can you provide the results generated by other methods, such as GAN, LAGAN, WGAN, RSGAN, Styleganv3?
  2. The ideas inspired from others should be cited, such as “principles of multi-scale visual perception”, “Dynamic feature gating”, etc.
  3. Is the generator architecture is designed by yourself?
  4. In Section 4.1, if eight categories, “including donut and center” should list all.
  5. In addition, you should list some necessary training hyperparameter.
  6. The data is from public dataset, you should cite them.

Author Response

Please see in attachment

Round 2

Reviewer 1 Report

Comments and Suggestions for Authors

thank you for the revisions and improvements to the manuscript. you have adequately addressed all of my comments and concerns. there are no other significant issues that I can identify

Reviewer 2 Report

Comments and Suggestions for Authors

All comments are addressed.